# Early Pregnancy Outcomes in Fresh Versus Deferred Embryo Transfer Cycles for Endometriosis-Associated Infertility: A Retrospective Cohort Study

**DOI:** 10.3390/jcm10020344

**Published:** 2021-01-18

**Authors:** Justin Tan, Maria Cerrillo, Maria Cruz, Gustavo Nardini Cecchino, Juan Antonio Garcia-Velasco

**Affiliations:** 1Department of Obstetrics and Gynaecology, Children’s and Women’s Hospital and Health Centre of British Columbia, University of British Columbia, Vancouver, BC V6H3N1, Canada; 2IVI-RMA, 28023 Madrid, Spain; Maria.Cerrillo@ivirma.com (M.C.); Maria.Cruz@ivirma.com (M.C.); gustavo.nardini@ivirma.com (G.N.C.); Juan.Garcia.Velasco@ivirma.com (J.A.G.-V.); 3Department of Gynecology and Obstetrics, Rey Juan Carlos University, 28922 Madrid, Spain

**Keywords:** endometriosis, freeze-all, endometrial receptivity, IVF

## Abstract

Given the estrogen-dependence associated with endometriosis, hyper-stimulation associated with assisted reproduction treatment may exacerbate the disease process and adversely affect endometrial receptivity and subsequent implantation. In this way, a freeze-all deferred embryo transfer (ET) approach may benefit patients with endometriosis, although controversy exists regarding the mechanism of endometriosis-associated infertility and benefits of deferred ET on endometrial receptivity. Hence, the purpose of this study was to compare in vitro fertilization (IVF) outcomes in women with endometriosis, diagnosed by histology, undergoing fresh versus deferred-ET after elective cryopreservation. Of the 728 women included, no significant differences were observed in baseline patient characteristics and response to gonadotrophin stimulation between fresh and deferred ET groups. Furthermore, no significant differences in implantation rate (49.7 vs. 49.9%, *p* = 0.73), clinical pregnancy rate (40.9 vs. 39.9%, *p* = 0.49), and miscarriage rate (9.4 vs. 9.9%, *p* = 0.63) were observed between fresh and deferred ET groups, respectively. Hence, contrary to previous studies, our results suggest that a deferred ET “freeze-all” IVF strategy does not improve early pregnancy outcomes among women with endometriosis. However, prospective studies are required to validate these findings and further insight into the etiology and pathogenesis of endometriosis-associated infertility are necessary to optimize IVF protocols in this population.

## 1. Introduction

Endometriosis is an estrogen-dependent disorder among women of reproductive age that affects their physical, mental, and social well-being. Therapeutic approaches are far from curative and the primary goal is oftentimes symptom management rather than eliminating the underlying disease process [1]. Assisted reproductive technologies (ART) represent the most successful means of achieving conception in endometriosis patients struggling with infertility and the increasingly widespread use of in vitro fertilization (IVF) has provided insights into possible mechanisms of endometriosis-related infertility.

Recently, there has been a growing trend towards a segmented, or freeze-all, approach to IVF [2,3]. This approach offers several potential advantages over fresh transfer cycles, including the ability to characterize embryo quality through preimplantation genetic testing (PGT-A), reduce the risk of ovarian hyperstimulation syndrome (OHSS), and reset the natural physiologic milieu for optimal implantation [4]. Indeed, several studies have shown improved reproductive outcomes with a freeze-all approach among high-responders and women with polycystic ovarian syndrome (PCOS) by avoiding embryo transfer in a hyper-stimulated milieu and optimizing the endometrial milieu with appropriate hormone replacement (HRT) in subsequent deferred embryo transfer (deferred-ET) cycles [5].

Given the estrogen-dependence associated with endometriosis, it is plausible that hyper-stimulation associated with assisted reproduction treatment may exacerbate the disease process and adversely affect endometrial receptivity and subsequent implantation; in this way, it has been proposed that a freeze-all deferred ET approach would mitigate these effects and improve IVF outcomes in patients with endometriosis [6,7]. However, controversy exists regarding the mechanism of endometriosis-associated infertility and the purported benefits of deferred ET with respect to endometrial receptivity [1,8,9]. Furthermore, data regarding freeze-all IVF outcomes in this particular population is scarce, with only a single study demonstrating higher cumulative pregnancy rates and lower miscarriage rates among endometriosis patients who underwent a deferred ET strategy [6]. Hence, the purpose of this study is to compare pregnancy outcomes in fresh and deferred ET cycles among women with endometriosis.

## 2. Experimental Section

### 2.1. Study Design

This retrospective cohort study was conducted at a tertiary care university-affiliated IVF center that included all infertile women undergoing their first IVF/ICSI cycle between January 2014 and July 2019. All stimulated IVF cycles were reviewed for possible inclusion and all patients were deemed eligible for IVF before initial treatment. A standard infertility evaluation was performed within 12 months of treatment, including baseline AMH, AFC, as well as transvaginal ultrasonography (TVUS) and assessment of the uterine cavity via hysterosalpingogram (HSG) or hysteroscopy. TSH and thyroid peroxidase antibody screening was also performed and thyroid hormone supplementation provided when indicated.

Two groups were compared: (1) a study group that included women with endometriosis who underwent deferred ET in a subsequent hormone replacement cycle for the first transfer attempt and (2) a “control” group comprised of women with endometriosis who underwent a fresh transfer cycle in their first IVF-ICSI attempt. In both groups, supernumerary embryos were frozen and transferred in subsequent cycles if successful pregnancy was not achieved. All procedures were approved by an Institutional Review Board on 16 January 2017 (1071-MAD-005-JG) at IVI Madrid.

### 2.2. Inclusion and Exclusion Criteria

Inclusion criteria for the cohort study were as follows: women with endometriosis-associated infertility undergoing their first single embryo transfer IVF-ICSI cycle, stimulation with GnRH antagonist protocol, age <43-years-old, and ≥1 embryo available for transfer. Exclusion criteria were as follows: GnRH agonist cycles, dual trigger protocols, severe male factor infertility requiring surgical retrieval of sperm, known parental genetic/monogenic diseases, and concurrent diagnosis of adenomyosis and/or other uterine or tubal anomalies. In all cases, endometriosis was diagnosed surgically by histologically proven biopsy and extent of disease reported by standardized ASRM classification [10]. The decision to undergo fresh or deferred ET with or without PGT-A was a joint discussion between patient and physician.

### 2.3. Ovarian Stimulation and Oocyte Retrieval

Ovarian stimulation was performed using a GnRH-antagonist protocol using a combination of recombinant FSH (Gonal F, Merck Serono, Spain) and highly purified urinary gonadotropins (Menopur, Ferring, Spain). Dosing regimens were chosen per physician discretion, considering patient specific factors such as age, BMI, and ovarian reserve tests (FSH, AMH and/or AFC). Gonadotropin was started on day 2–3 of the menstrual cycle. Fixed start of antagonist (Orgalutran, MSD, Spain) was initiated on day 6 of stimulation and continued until the day of the trigger. In the deferred ET group, final oocyte maturation was at the discretion of the treating physician and achieved by administering either recombinant hCG (Ovitrelle 250 μg, Merck Serono, Spain) or a single bolus of GnRH-a (Decapeptyl 0.2 mg, Ferring, Switzerland) when at least three follicles were ≥17 mm in mean diameter. In the fresh-ET group, final oocyte maturation was achieved using hCG trigger exclusively. Oocyte retrieval was performed 36 h after triggering by TVUS-guided aspiration.

#### 2.3.1. Oocyte Insemination

After a period of abstinence ranging from 3–5 days, semen samples were collected by masturbation and standard sperm parameters were used to determine insemination by conventional IVF or injection by ICSI. Subsequently, fertilization was assessed by the presence of two pronuclei (2PN) and two polar bodies at 17–18 h post-insemination.

#### 2.3.2. Embryo Culture and Preimplantation Genetic Testing (PGT-A)

In fresh transfer cycles, embryos were cultured in a single droplet of culture media (Global Plus, Life Global). Based on validated grading criteria [11], embryo morphology was evaluated on day 5 and considered eligible for transfer with a viable grade. In PGT-A cycles, culture was continued in individual droplets (Global Plus, Life Global) until day 5 or 6 and trophectoderm (TE) biopsy performed on blastocysts with a viable grade. Micromanipulation and biopsy were performed in HEPES supplemented with 20% serum with the use of laser pulses to release five to ten TE cells for analysis. The precise vitrification and thawing protocol in our group has been reported in detail previously [12]. Blastocysts were warmed on the day of transfer.

#### 2.3.3. Endometrial Preparation and ET

In fresh ET cycles, progesterone treatment (Progeffik 200 mg BID) was initiated the day after oocyte retrieval. Generally, a single or double embryo transfer was performed based on a morphologic evaluation. In deferred ET cycles, 6 mg oral estradiol (Estradiol Meriestra, Sandoz, Spain) was administered beginning on day 2 or 3 following menses. Transvaginal sonography (TVUS) was used to assess the pattern and thickness of the endometrium approximately 12 days after menses, and progesterone (Progeffik 400 mg BID) was administered when a trilaminar pattern was achieved with a thickness ≥ 8 mm in fresh ET cycles and ≥ 7 mm in deferred-ET cycles. Generally, embryo transfer was performed after 5 days of progesterone treatment and luteal support continued until 12 weeks gestation.

#### 2.3.4. Main Outcomes and Statistical Analysis

Implantation rate (IR), clinical pregnancy rate (CPR) and miscarriage rate (MR) were the primary outcomes of interest. IR was calculated based on the number of gestational sacs observed at echographic screening between 3–5 weeks gestation as a proportion of the number embryos transferred. CPR was defined by the presence of a viable fetal heart rate and crown rump length (CRL) ≥ 10 mm by U/S performed between 7–9 weeks’ gestation, while MR included both biochemical and clinical losses.

Data from clinical outcomes are presented as descriptive statistics (means with standard deviations (SDs)). One-way analysis of variance for continuous variables and the chi-squared test for categorical data were used for data analysis. Statistical analysis was performed with the Statistical Package for Social Sciences, version 20.0 (SPSS, IBM Corporation, NY, USA). In all cases, statistical significance was defined as *p* < 0.05.

## 3. Results

### 3.1. Baseline Patient Characteristics

Among 728 unique patients with endometriosis, 339 women underwent fresh ET and 389 underwent deferred ET. As shown in Table 1, no significant differences in age, BMI, duration of infertility, and baseline AMH (3.4 ± 1.8 vs. 4.3 ± 2.2, *p* = 0.18) and AFC (7.6 ± 2.7 vs. 8.9 ± 1.1) were observed between fresh and deferred ET groups, respectively. With respect to causes of infertility, similar rates of male factor and tubal factor infertility were observed, while the total doses of gonadotropin used were also similar fresh and deferred ET groups (2690± 141 vs. 2635 ± 185 IU, *p* = 0.34, respectively).

### 3.2. IVF-ICSI Outcomes in Non-PGT-A Cycles

As shown in Table 2, the number of oocytes (7.4 ± 0.3 vs. 8.2 ± 0.8, *p* = 0.57) and MII oocytes retrieved were similar between fresh and deferred ET groups (5.6 ± 0.6 vs. 6.0 ± 0.7, *p* = 0.45), respectively. The fertilization rate (75.2 vs. 75.1%, *p* = 0.81) was also similar between fresh and deferred ET groups, respectively. Furthermore, no significant differences were observed with respect to implantation (49.7 vs. 49.9%, *p* = 0.73), clinical pregnancy rates (40.9 vs. 39.9%, *p* = 0.49), and miscarriage rates (9.4 vs. 9.9%, *p* = 0.63) between fresh and deferred ET groups, respectively.

### 3.3. Age Stratified IVF-ICSI Outcomes (<35, 35–38, >38) in Non-PGT-A Cycles

As shown in Table 3, ovarian response to stimulation decreased with increasing maternal age as fewer oocytes were retrieved among women >38-years-old compared to women <35-years-old in both fresh and deferred ET groups. Similarly, implantation and clinical pregnancy rates decreased with increasing maternal age, while miscarriage rates were similar across all age groups.

In both fresh and deferred ET cycles, implantation rates were comparable in the <35-year-old (55.8 vs. 54.9%, *p* = 0.45), 35–37-year-old (46.8 vs. 48.2%, *p* = 0.45), and >38-year-old (42.3 vs. 43.7%, *p* = 0.27) groups. Similarly, clinical pregnancy rates and miscarriage rates were also similar between fresh and deferred ET groups across all age categories.

### 3.4. PGT-A Cycles

As shown in Table 4, a comparison of PGT-A cycles demonstrates a similar number of oocytes retrieved (9.1 ± 1.3 vs. 8.6 ± 1.6, *p* = 0.46), MII oocytes retrieved (8.2 ± 0.7 vs. 7.5 ± 0.9, *p* = 0.14), and fertilization rates (74.2 vs. 72.6%, *p* = 0.28) between fresh and deferred ET cycles, respectively. With respect to IVF-ICSI outcomes after euploid embryo transfer, implantation (45.2 vs. 46.6%, *p* = 0.38) and clinical pregnancy (43.4 vs. 45.1%, *p* = 0.26) rates were similar in fresh and deferred ET cycles, respectively. Miscarriage rates were also similar between groups (4.6 vs. 5.2%, *p* = 0.12) and consistently lower than the non-PGT-A study cohort (9.4 and 9.9% in fresh and deferred-ET cycles, respectively), likely owing to the fact that only euploid embryos were transferred.

## 4. Discussion

Based on the results of this study, women with endometriosis-associated infertility can expect similar pregnancy rates in fresh and deferred ET cycles across all maternal age-groups. Particularly in PGT-A cycles, whereby the influence of ovarian factors is greatly reduced by eliminating the influence of embryo aneuploidy on pregnancy outcomes, a freeze-all strategy does not appear to confer improved implantation or pregnancy outcomes among women with endometriosis-associated infertility. The primary strength of this study is its large sample size and similar baseline patient characteristics with respect to duration of infertility, baseline ovarian reserve, and ovarian response to stimulation, thereby reinforcing the comparability between groups. Furthermore, all included patients attended a single tertiary care institution, which minimizes practice variation between physicians and the inclusion of a subgroup comparing only embryos screened by PGT-A significantly reduces the influence of embryo quality on comparing ET outcomes after IVF. As the influence of embryo aneuploidy is removed, PGT-A cycle comparisons are the most reliable for comparing the impact of fresh vs. deferred ET outcomes on endometrial receptivity defects associated with endometriosis.

Conversely, the major limitation of this study is its retrospective design. Although all patients attended a single fertility clinic, the decision to choose a GnRHa or hCG trigger may have been influenced by each physicians’ intrinsic bias and subjective suspicion of OHSS risk, which increases heterogeneity and likelihood of confounding. However, it is important to note that the baseline characteristics of patients in both groups were similar, which supports the validity of such a comparison. Furthermore, diagnostic details beyond ASRM classification such as location of endometriosis, duration since diagnosis, and concurrent use of co-medication were not reported. Finally, we recognize that our study reports early pregnancy outcomes while live birth rate (LBR) remains the gold standard and most clinically meaningful outcome of interest. Given these limitations, further prospective and randomized control studies are required to confirm these findings and should include women with a gold standard laparoscopic diagnosis of endometriosis and detailed assessment of the location and extent of disease.

Endometrial receptivity is mediated by a complex process involving endocrine, paracrine, and autocrine factors. While numerous endometrial markers have been proposed to characterize this complex signature of optimal receptivity, none have been widely incorporated into clinical practice due to their poor ability to predict pregnancy [13,14]. Nevertheless, dysregulation of genes involved in angiogenesis, steroidogenesis, and inflammation at the level of the endometrium have been described in women with endometriosis [15,16,17,18]. Hence, it remains biologically plausible that endometriosis-associated endometrial receptivity defects could be exacerbated by exposure to high levels of steroids in an ART cycle, thereby supporting a “freeze-all” deferred ET IVF strategy. Indeed, a recent study by Bourdon et al. [6] found that women who underwent deferred ET experienced a significantly higher live birth rate per cycle (28.9 vs. 15.6%, *p* = 0.025) and cumulative pregnancy rate (43 vs. 29.6%, *p* = 0.047) compared to women who underwent a fresh embryo transfer. However, the authors note several limitations to their conclusions, including significant differences in study populations and stimulation protocols between groups. In particular, women in the fresh ET group demonstrated a less efficient response to stimulation overall and the mean number of embryos transferred was significantly higher compared to the deferred ET group (2.1 ± 0.9 vs. 1.7 ± 0.9, *p* < 0.001). Conversely, our study benefits from a larger sample size and greater similarity in baseline patient characteristics, stimulation protocols used, and stimulation response between groups. Furthermore, the inclusion of a subgroup of PGT cycles further minimizes the potential confounding associated with differences in embryo quality between groups.

Although our results suggest that a deferred ET strategy cannot mitigate the purported endometrial receptivity defects associated with endometriosis, an alternative hypothesis may be related to the pathogenesis of endometriosis-associated infertility; namely, a primary defect residing in the ovary and impaired oocyte quality as opposed to defective implantation. While several studies have reported changes to the eutopic endometrium in natural cycles, whether those changes adversely affect endometrial receptivity or clinical outcomes in stimulated and hormone replacement cycles has not been demonstrated. In fact, a growing body of molecular and clinical evidence suggests that the mechanism of endometriosis-associated infertility may not be caused by defects in endometrial receptivity [1].

Several studies have found altered gene expression patterns [14,19,20,21,22,23] and dysregulated steroid hormone pathways [19,24] in the eutopic endometrium of women with endometriosis. However, the relative implications with respect to endometrial receptivity are less conclusive and clinical studies among donor oocyte cycles offer the clearest evidence with respect to the origin of endometriosis-associated infertility. Indeed, [25] observed that women with endometriosis who received donor oocytes achieved similar pregnancy outcomes compared to women with other causes of infertility, while donor recipients who received oocytes from women with endometriosis experienced significantly lower implantation rates (*p* < 0.05), thereby suggesting an oocyte-driven cause of endometriosis-associated infertility instead of a defect in endometrial receptivity. To further minimize the effects of differences in oocyte quality, Diaz et al. [26] conducted a matched case-control study in which healthy oocyte donors shared their oocytes equally between women with stage III-V endometriosis and healthy controls. Overall, women with stage III-IV endometriosis demonstrated similar implantation (14.8 vs. 16.0%), ongoing pregnancy (40.0 vs. 45.5%), live birth (28 vs. 27.2%), and miscarriage rates (28.0 vs. 27.2%) compared to the healthy control group, respectively, thereby further suggesting that severe endometriosis does not impair endometrial receptivity or diminish the implantation potential of donated oocytes in HRT cycles.

Basic science studies have also contested the theory of defective implantation capacity as the probable cause of endometriosis-associated infertility. In particular, proteomic and transcriptomic studies have enabled a molecular phenotyping of the endometrium beyond the single-molecule approach of identifying candidate biomarkers of implantation using a molecular tool that evaluates the expression of 238 genes related to endometrial receptivity [27,28]. Garcia-Velasco et al. [29] used the clinically validated endometrial receptivity array (ERA) to evaluate the receptivity of patients with different stages of endometriosis as well as healthy controls. Interestingly, comparisons between endometriosis stages demonstrated no significant differences in the expression patterns of genes related to endometrial receptivity, thereby suggesting that endometrial receptivity is similar between women with and without endometriosis, as well as across different stages of endometriosis.

In contrast, novel diagnostic techniques for evaluating oocyte quality and developmental competence have reinforced the concept of an ovarian cause of endometriosis-associated infertility. Using transmission electron microscopy (TEM), Xu et al. [30] noted an abnormal mitochondrial structure, decreased mitochondria mass, reduced mitochondrial DNA copy number compared to oocytes from healthy controls, suggesting an adverse effect of endometriosis on gamete folliculogenesis. Furthermore, Boynukalin et al. [31] used time-lapse imaging to evaluate the effect of endometriosis on early cell cycle events and noted altered relative morphokinetics that suggested poorer embryo quality compared to developing embryos from healthy controls [32]. Taken together, these findings may explain the reduced numbers of oocytes retrieved and the lower overall fertilization rates among women with endometriosis compared to women with tubal factor infertility (RR 0.97, CI 0.96 to 0.98, *p* = 0.0001) as observed in recent population-based studies using data from the Society for Assisted Reproductive Technology (SART) [33]. Along with similar results in both meta-analyses [34,35,36,37] and recent prospective studies [38,39], these findings support the theory that mechanisms of endometriosis-associated infertility are unrelated to defects in endometrial receptivity but rather diminished oocyte and embryo competence. In absolute terms, however, it is also important to recognize that patients with a diagnosis of endometriosis presenting for IVF often have at least one other infertility diagnosis, and among those with endometriosis in isolation, pregnancy outcomes are similar or slightly higher compared to women with other infertility diagnoses [33].

## 5. Conclusions

Based on the results of this study, a deferred-ET “freeze-all” IVF strategy does not improve pregnancy outcomes among women with endometriosis, but may increase time to pregnancy and add unnecessary interventions, such as embryo freezing. Further randomized prospective studies are required to validate these findings and future refinements to optimize IVF protocols will require further insight into the etiology and pathogenesis of endometriosis-associated infertility. Endometriosis management in the context of infertility should likely be based on a personalized approach to ART treatment.

## Figures and Tables

**Table 1 jcm-10-00344-t001:** Baseline patient characteristics.

BASELINE CHARACTERISTICS	FRESH-ET (*n* = 339)	DEFERRED-ET (*n* = 389)	*p*-VALUE
AGE (YEARS) *	35.5 ± 0.2	35.9 ± 0.3	0.34
BMI (KG/M2) *	22.2 ± 0.1	22.4 ± 0.2	0.51
DURATION OF INFERTILITY (YEARS) *	2.8 ± 0.3	2.9 ± 0.3	0.31
OVARIAN RESERVE			
AMH (NG/ML) *	3.4 ± 1.8	4.3 ± 2.2	0.18
AFC *	7.6 ± 2.7	8.9 ± 1.1	0.16
GRAVIDITY			0.12
0	91.6%	91.2%	
≥1	8.4%	8.8%	
PARITY			0.25
0	94.8%	94.3%	
≥1	5.2%	5.7%	
ASSOCIATED MALE INFERTILITY	3.2%	2.9%	0.37
ENDOMETRIOSIS I-II STAGE	37.6%	39.7%	0.24
ENDOMETRIOSIS III-IV STAGE	62.4%	60.3%	0.17
TOTAL DOSES OF FSH (IU) *	1686 ± 46	1673 ± 64	0.14
TOTAL DOSES OF HMG (IU) *	1004 ± 95	962 ± 121	0.47

* values presented as mean ± SD. *n* = number of study participants.

**Table 2 jcm-10-00344-t002:** IVF-ICSI outcomes in non-PGT-A cycles.

CYCLE OUTCOMES	FRESH-ET (*n* = 287)	DEFERRED-ET (*n* = 306)	*p*-VALUE
RETRIEVED OOCYTES	7.4 ± 0.3	8.2 ± 0.8	0.57
MII OOCYTES	5.6 ± 0.6	6.0 ± 0.7	0.45
FERTILIZATION RATE (%)	75.2%	75.1%	0.81
IMPLANTATION RATE	49.7%	49.9%	0.73
CLINICAL PREGNANCY RATE	40.9%	39.9%	0.49
MISCARRIAGE RATE	9.4%	9.9%	0.63

**Table 3 jcm-10-00344-t003:** Age-stratified IVF-ICSI outcomes in non-PGT-A Cycles.

CYCLE OUTCOMES	FRESH-ET	DEFERRED-ET	*p*-VALUE
**<35 YEARS-OLD**	***n* = 124**	***n* = 128**	
RETRIEVED OOCYTES	10.1 ± 0.7	10.5 ± 1.2	0.59
MII OOCYTES	7.3 ± 0.5	8.1 ± 0.5	0.61
FERTILIZATION RATE (%)	78.9%	79.6%	0.52
IMPLANTATION RATE	55.8%	54.9%	0.45
CLINICAL PREGNANCY RATE	48.6%	44.2%	0.08
MISCARRIAGE RATE	7.3%	8.4%	0.12
**35–37 YEARS-OLD**	***n* = 99**	***n* = 101**	
RETRIEVED OOCYTES	8.8 ± 0.7	10.2 ± 1.3	0.48
MII OOCYTES	6.7 ± 0.5	7.8 ± 0.3	0.68
FERTILIZATION RATE (%)	74.8%	73.9%	0.73
IMPLANTATION RATE	46.8%	48.2%	0.45
CLINICAL PREGNANCY RATE	37.2%	38.7%	0.48
MISCARRIAGE RATE	8.2%	9.8%	0.16
**>38 YEARS-OLD**	***n* = 64**	***n* = 77**	
RETRIEVED OOCYTES	5.8 ± 0.6	6.7 ± 1.2	0.41
MII OOCYTES	4.1 ± 0.9	4.6 ± 0.8	0.24
FERTILIZATION RATE (%)	68.5%	69.2%	0.32
IMPLANTATION RATE	42.3%	43.7%	0.27
CLINICAL PREGNANCY RATE	31.6%	34.5%	0.19
MISCARRIAGE RATE	10.4%	10.8%	0.26

**Table 4 jcm-10-00344-t004:** IVF-ICSI Outcomes in PGT-A cycles.

CYCLE OUTCOMES	FRESH TRANSFER (*n* = 52)	DEFERRED TRANSFER (*n* = 83)	*p*-VALUE
**RETRIEVED OOCYTES**	9.1 ± 1.3	8.6 ± 1.6	0.46
**MII OOCYTES**	8.2 ± 0.7	7.5 ± 0.9	0.14
**FERTILIZATION RATE (%)**	74.2%	72.6%	0.28
**IMPLANTATION RATE**	45.2%	46.6%	0.38
**CLINICAL PREGNANCY RATE**	43.4%	45.1%	0.26
**MISCARRIAGE RATE**	4.6%	5.2%	0.12

## Data Availability

The data presented in this study are available on request from the corresponding author.

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
