# Peer review of "Early Pregnancy Outcomes in Fresh Versus Deferred Embryo Transfer Cycles for Endometriosis-Associated Infertility: A Retrospective Cohort Study"

_jcm, 2021, doi:10.3390/jcm10020344_

Round 1

Reviewer 1 Report

I think there are still errors in the results:

- in paragraph 3.1 authors write: “Among 863 unique patients with endometriosis, 391 women underwent fresh-ET and 472 underwent deferred-ET”, but in table 1 they write: FRESH-ET (N=287) and DEFERRED-ET (N=316).

- in paragraph 3.2 some values ​​do not correspond with what is reported in table 2 (fertilization rate, p of clinical pregnancy rates, p of miscarriage rates)

- the total numbers of patients and of patients who underwent cycles with and without PGT-A also appear to be different: in table 2 (IVF-ICSI outcomes in non-PGT-A cycles) the numbers of patients are 287 for FRESH-ET versus 306 for DEFERRED-ET and in table 3 (IVF-ICSI Outcomes in PGT-A cycles) 52 versus 83. The sum of the patients does not correspond to the total number of patients reported in table 1 FRESH-ET (N=287) and DEFERRED-ET (N=316).

Author Response

I think there are still errors in the results:

- in paragraph 3.1 authors write: “Among 863 unique patients with endometriosis, 391 women underwent fresh-ET and 472 underwent deferred-ET”, but in table 1 they write: FRESH-ET (N=287) and DEFERRED-ET (N=316).

Response: Agreed and amended. The values in the tables are correct but were not fully transcribed to the manuscript text. Our revised analysis includes 728 patients as the sum of 339 patients in the fresh-ET group (287 non-PGTA and 52 PGTA) and 389 patients in the deferred-ET group (306 non-PGTA and 83 PGT-A). Our sincerest apologies for this oversight.

- in paragraph 3.2 some values ​​do not correspond with what is reported in table 2 (fertilization rate, p of clinical pregnancy rates, p of miscarriage rates)

Response: Agreed and amended. As above, the values in table 2 are correct and amended in the manuscript text.  

- the total numbers of patients and of patients who underwent cycles with and without PGT-A also appear to be different: in table 2 (IVF-ICSI outcomes in non-PGT-A cycles) the numbers of patients are 287 for FRESH-ET versus 306 for DEFERRED-ET and in table 3 (IVF-ICSI Outcomes in PGT-A cycles) 52 versus 83. The sum of the patients does not correspond to the total number of patients reported in table 1 FRESH-ET (N=287) and DEFERRED-ET (N=316).

Response: Agreed and amended.

Reviewer 2 Report

This revised version of the paper is somewhat improved, but there are still some discrepancies needing to be resolved.

The authors state in the accompanying letter that endometriosis diagnosis was based on ASRM criteria. I fail to find this in the methods section and, the mention of ultrasound diagnosis inclusion is still present in the Abstract. Without a further detailed explanation of which criteria were used and how they were implemented (did the center collect histological evidence on all patients or was this as per the patient’s history?)  and a section on further description of diagnostic details (stage, duration since diagnosis, comedication used etc), this paper falls short of a minimally acceptable patient material description. Importantly, was endometriosis the only infertility diagnosis in the couples?

Another main comment relates to the physician’s allocation of patients to receive either a segmented cycle or fresh embryo transfer. This decision was understandably based on the treating physician´s choice of type of oocyte maturation trigger (hCG or GnRH agonist). Given there might have been changes in clinical routine over recent years, and as the study included patients from a period of more than 5 years, it would seem appropriate to describe this in more detail, for instance were there any differences between different year-to -year periods? Inherently, the possible selection bias is embedded in the retrospective design, which is acknowledged by the authors, but the patient material deserves further detailed description to enable any such (possible) differences better.  This is also mentioned in the accompanying letter, but this would benefit if being eluted to in text of the manuscript, as it is very surprising to note there were no differences between the treatment group regarding ovarian reserve measurements, oocyte numbers, etc. How can this be explained?

Finally, I am struggling to understand why this study does not report on live birth outcome. Stating that the dataset ´does not include live birth rates´ seems as overly simplistic explanation as I would assume this renowned center routinely collect live birth data.

Minor comments:

Table 1 has data on dosages of LH. Assumingly, this reflects use of recLH (why?) and this is not mentioned in the ovarian stimulation section. Please add and explain. The Discussion section related to the previous study of Bourdain et al and would benefit of providing their relative (%) or actual differences (numerical) rather than a p-value.

The Discussion section related to existence of an endometriosis related oocyte factor (lines 242 – 271) is overly long and can be markedly shortened. Instead of only referencing as possible – albeit small - impact of oocyte fertilization rate, in fact the SART data paper from 2016 (Senapati) highlights that most couples with a diagnosis of endometriosis presenting for IVF will have at least one other infertility diagnosis, and that in contrast, for the minority of women who have endometriosis in isolation, the live birth rate is similar or slightly higher compared with other infertility diagnoses. As such, this can suggestively also be mentioned in the Discussion to balance the views on impact of endometriosis on fecundity and IVF outcomes.

Line 275. It needs to be emphasized the need to further randomized prospective studies

Author Response

This revised version of the paper is somewhat improved, but there are still some discrepancies needing to be resolved.

The authors state in the accompanying letter that endometriosis diagnosis was based on ASRM criteria. I fail to find this in the methods section and, the mention of ultrasound diagnosis inclusion is still present in the Abstract.

Response: For all patients, Endometriosis was diagnosed surgically by histologically proven biopsy. Since we do not perform surgery at our center, this information was provided by surgical and histopathology reports from community gynecologists.

Regarding ASRM criteria, our clinical database for patients with surgically-confirmed endometriosis categorizes the extent of disease as either ASRM I-II or ASRM III-IV and this is now included in revised table 1. This data was actually included in a prior version but previous reviewers suggested that it be removed given that, at the time, it only pertained to a subset of our study group when we also included patients for whom endometriosis was diagnosed by history and/or ultrasound. Given that we have now limited our study group to those patients with a surgical diagnosis, we agree that it is now appropriate to include this data. The methods and abstract have also been updated to reflect this.

Without a further detailed explanation of which criteria were used and how they were implemented (did the center collect histological evidence on all patients or was this as per the patient’s history?)  and a section on further description of diagnostic details (stage, duration since diagnosis, comedication used etc), this paper falls short of a minimally acceptable patient material description. Importantly, was endometriosis the only infertility diagnosis in the couples?

Response: As above, all surgeries were performed by community gynecologists and we were provided reports regarding histological confirmation and intra-operative ASRM classification.

Regarding concurrent diagnoses, our study criteria excluded patients with adenomyosis and/or other uterine or tubal anomalies as well as those patients with known parental genetic/monogenic diseases. Severe male factor infertility requiring surgical retrieval of sperm was also an exclusion criteria, while mild male factor infertility was not excluded but the prevalence was similar between groups.

The duration since diagnosis, use of co-medication, and specific location of disease was not recorded in our database. Although this may have an impact on ovarian reserve (particularly in the case of endometriomas) or anatomic distortion (tubal disease), it is important to note that ovarian reserve markers were similar between study groups and patients with tubal anomalies / block were excluded. Nevertheless, we agree that this is a limitation of our study and have amended our discussion to highlight this point.

Another main comment relates to the physician’s allocation of patients to receive either a segmented cycle or fresh embryo transfer. This decision was understandably based on the treating physician´s choice of type of oocyte maturation trigger (hCG or GnRH agonist). Given there might have been changes in clinical routine over recent years, and as the study included patients from a period of more than 5 years, it would seem appropriate to describe this in more detail, for instance were there any differences between different year-to -year periods? Inherently, the possible selection bias is embedded in the retrospective design, which is acknowledged by the authors, but the patient material deserves further detailed description to enable any such (possible) differences better.  This is also mentioned in the accompanying letter, but this would benefit if being eluted to in text of the manuscript, as it is very surprising to note there were no differences between the treatment group regarding ovarian reserve measurements, oocyte numbers, etc. How can this be explained?

Response: Regarding the purported temporal bias, we perform regular audits of our clinical practices and patient outcomes and have not found any significant temporal trends besides a slight increase in the proportion of freeze-all and PGT-A cycles. Furthermore, clinical and laboratory staff were consistent throughout the study period hence any major changes in routines are unlikely. Conversely, the length of study period was necessary in order to obtain a meaningful sample size for analysis. Nevertheless, we agree that this is an unavoidable limitation of retrospective studies and acknowledge this in the discussion.

Finally, I am struggling to understand why this study does not report on live birth outcome. Stating that the dataset ´does not include live birth rates´ seems as overly simplistic explanation as I would assume this renowned center routinely collect live birth data.

Response: As we mentioned in our previous revision, we agree that live birth outcomes remain the gold standard primary outcome of interest, but this data is not readily available in our clinic database. Live birth data falls outside the scope of our original IRB application as it is collected by a national registry that is not directly connected to our IVF dataset. With respect to the value of early pregnancy outcomes, Arce et al. (2005)1 discuss the validity of different outcomes of interest in ART studies and support the use of earlier clinical endpoints that correlate closely with LBR as long as it reasonably supports the study question. Given that the primary purpose of our study was to assess the impact of endometriosis on endometrial receptivity, which affects early pregnancy outcomes, we feel that implantation and clinical pregnancy rates serve as reasonable primary outcomes to evaluate this study question.

Minor comments:

Table 1 has data on dosages of LH. Assumingly, this reflects use of recLH (why?) and this is not mentioned in the ovarian stimulation section. Please add and explain.

Response: We apologize for this error. Although some patients received rLH during at our clinic during this study period, this minority of patients were not included in our revised study groups so LH dosages have been removed from Table 1. All included patients received rFSH and Menopur only as per the methods section and our total study size is now 728 (previously 863).

The Discussion section related to the previous study of Bourdain et al and would benefit of providing their relative (%) or actual differences (numerical) rather than a p-value.

Response: Agreed and amended with relative differences.

The Discussion section related to existence of an endometriosis related oocyte factor (lines 242 – 271) is overly long and can be markedly shortened. Instead of only referencing as possible – albeit small - impact of oocyte fertilization rate, in fact the SART data paper from 2016 (Senapati) highlights that most couples with a diagnosis of endometriosis presenting for IVF will have at least one other infertility diagnosis, and that in contrast, for the minority of women who have endometriosis in isolation, the live birth rate is similar or slightly higher compared with other infertility diagnoses. As such, this can suggestively also be mentioned in the Discussion to balance the views on impact of endometriosis on fecundity and IVF outcomes.

Response: The last two paragraphs have been shortened and we have further elaborated on the findings by Senapati et al. as suggested by the reviewer.

Line 275. It needs to be emphasized the need to further randomized prospective studies

Response: This has been amended to specify randomized prospective studies.

References:

Arce, J. C., et al. (2005). "Resolving methodological and clinical issues in the design of efficacy trials in assisted reproductive technologies: a mini-review." Hum Reprod 20(7): 1757-1771.

Round 2

Reviewer 2 Report

The Authors have provided acceptable answers to the queries raised and revised the paper accordingly. 

This manuscript is a resubmission of an earlier submission. The following is a list of the peer review reports and author responses from that submission.

Round 1

Reviewer 1 Report

This is a retrospective cohort study comparing the results of fresh embryo transfers to a freeze all strategy with deferred embryo transfer in pts with endometriosis associated infertility. The theory behind this strartegy is that since endometriosis is an estrogen dependent entity, the high E2 levels obtained during COS may have a detrimental effect on the secretory endometrium and inferior implantation rates compared with deferred transfers in estrogen/progesterone substituted cycles. Unfortunately there are several flaws in the paper which make the conclusions dubious. The diagnosis of endometriosis in this study sample is inadequate since TVUS can only diagnose DIE/endometrioma. Only 20% had previously undergone surgery for endometriosis and in Table 1, only 40% in both cohorts had their endometriosis staged. What about the rest of the pts? Who made the decision to perform a freeze all cycle and on which grounds? There is a highly skewed distribution of the PGT-A cycles, and were these cycles performed on pts with endometriosis or not?  It is stated that the endometrium and its receptivity in deferred cycles is optimized with HRT. 6 mg of estradiol was given daily until a endometrial diameter of 6 mm which is lower than most clinics would accept. Is this an optimal stimulation of the endometrium, and what is the serum conc. of E2 during such treatment?. The miscarriage rate in the derferred cycles group was much higher than in the fresh transfer group, so perhaps not so optimal endometrium after all? The title is inadequate since  the study only reports on implantation rate , clinical pregnancy rate and miscarrage rate in the first trimester. There  are some words missing.

Reviewer 2 Report

The paper is very clear, but I think there are some errors in the results:

  1. authors write: “In the deferred-ET group, a slightly higher proportion of women had a history of previous surgery for endometriosis (20.8 vs. 16.3%, p=0.048)” but in table 1 they show different results: 20.8 vs 18.3 (p=0.148).
  2. The authors compare 467 women underwent fresh-ET and 548 underwent 127 deferred-ET but in table 3 the sum of patients who underwent fresh ET is 463 while deferred ET is 555.
  3. The authors correctly divide patients in PGT-A cycles (table 4: 76 and 179 patients) and non PGT-A cycles (table 2: 467 and 548 patients). In tables 1 they describe the whole population of 467 and 548 patients, that corresponds to non PGT-A cycles: patients that underwent PGT_A cycles are not included in table 1?

Reviewer 3 Report

I have read the authors' manuscript with interest. The investigators report on ART outcome of 1015 infertile women who underwent a first IVF/ICSI at a single center during a study period of 5 ½+ years. Based on information retrieved form a prospectively maintained database the authors have attempted to compare outcome of fresh embryo transfer cycles and so-called deferred embryo transfer in women with endometriosis.

Aside from the main comment that it is a retrospective cohort analysis with its inherent weaknesses, to my opinion, the study design and interpretation of data is flawed in several ways. Most importantly, it is not adequately described as to the how the subjects were allotted to the two different treatment regimens and thereby this introduces the possibly of selection bias.  Moreover, the study endpoints are described as implantation rate and clinical pregnancy rate at gestational weeks 7-9 (why such a long period?), rather than live birth rate (LBR). This is surprising, since LBR has been recognized as the gold standard for many years now, and arguably it would have been easy to include given the retrospective study design. It is also interesting to note that the clinic chooses to transfer 2 blastocysts (or was this indeed the case? I cannot find any description of embryo stage and quality, hence I inferred blastocyst transfer from 5 days of progesterone for luteal phase support being mentioned) in more than half of patients despite the inherent risk of multiple pregnancies. This should, to my opinion, also be commented as it goes against most clinical guideline recommendations.  

The decision to transfer either one or two embryos is also not adequately described, and even if the number in the respective groups turned out similar, there is no multivariate testing methods applied to describe possible other influencing factors. Likewise, a subgroup of patients underwent PGT-A, the reason for the is left for speculation as it is not mentioned, and the methodology used is not disclosed. I am also somewhat surprised to see a group of fresh transfer PGT-A patients being included with embryo transfer on D5/6, since most units today would opt for freezing biopsied embryos while awaiting the results of the genetic analysis and transfer selected embryo in s subsequent (or as by terminology used herein, ´deferred´) cycle.

The discussion is overly elaborate and auspiciously directed to convince the reader of a possible oocyte effect on endometriosis, although this was not the aim of the study. To the latter, it is also surprising there is no mention of the recently published large randomized studies on fresh versus frozen embryo transfers and associated conjectures. Moreover, the clinic is apparently using only HRT programmed cycle and there is no mentioning of natural or natural-modified cycles, perhaps this regimen would also be worthwhile to include a comparison of?

Therefore, overall, this study does not substantively add to the prior studies and the outcome of such complex study groups will necessitate better described methodology and as the authors recognize prospective randomized design with sharpened criteria than herein.